# Continuous In Vitro Culture of *Babesia duncani* in a Serum-Free Medium

**DOI:** 10.3390/cells12030482

**Published:** 2023-02-02

**Authors:** Weijun Jiang, Sen Wang, Dongfang Li, Yajun Zhang, Wanxin Luo, Junlong Zhao, Lan He

**Affiliations:** 1State Key Laboratory of Agricultural Microbiology, College of Veterinary Medicine, Huazhong Agricultural University, Wuhan 430070, China; 2Key Laboratory of Preventive Veterinary Medicine in Hubei Province, College of Animal Science and Technology, Huazhong Agricultural University, Wuhan 430070, China; 3Key Laboratory of Animal Epidemical Disease and Infectious Zoonoses, Ministry of Agriculture, Huazhong Agricultural University, Wuhan 430070, China

**Keywords:** babesiosis, *Babesia duncani*, serum-free culture, animal component-free medium, AlbuMax^TM^ I, CD lipid mixture

## Abstract

Human babesiosis is an emerging tick-borne disease, caused by haemoprotozoa genus of *Babesia*. Cases of transfusion-transmitted and naturally acquired *Babesia* infection have been reported worldwide in recent years and causing a serious public health problem. *Babesia duncani* is one of the important pathogens of human babesiosis, which seriously endangers human health. The in vitro culture systems of *B. duncani* have been previously established, and it requires fetal bovine serum (FBS) to support long-term proliferation. However, there are no studies on serum-free in vitro culture of *B. duncani*. In this study, we reported that *B. duncani* achieved long-term serum-free culture in VP-SFM AGT^TM^ (VP-SFM) supplemented with AlbuMax^TM^ I. The effect of adding different dilutions of AlbuMax^TM^ I to VP-SFM showed that 2 mg/mL AlbuMax^TM^ I had the best *B. duncani* growth curve with a maximum percentage of parasitized erythrocytes (PPE) of over 40%, and it can be used for long-term in vitro culture of *B. duncani*. However, the commonly used 20% serum-supplemented medium only achieves 20% PPE. Clearly, VP-SFM with 2 mg/mL AlbuMax^TM^ I (VP-SFMA) is more suitable for the in vitro proliferation of *B. duncani.* VP-SFM supplemented with CD lipid mixture was also tested, and the results showed it could support the parasite growth at 1:100 dilution with the highest PPE of 40%, which is similar to that of 2 mg/mL AlbuMax^TM^ I. However, the CD lipid mixture was only able to support the in vitro culture of *B. duncani* for 8 generations, while VP-SFMA could be used for long-term culture. To test the pathogenicity, the VP-SFMA cultured *B. duncani* was also subjected to hamster infection. Results showed that the hamster developed dyspnea and chills on day 7 with 30% PPE before treatment, which is similar to the symptoms with un-cultured *B. duncani*. This study develops a unique and reliable basis for further understanding of the physiological mechanisms, growth characteristics, and pathogenesis of babesiosis, and provides good laboratory material for the development of drugs or vaccines for human babesiosis and possibly other parasitic diseases.

## 1. Introduction

Human babesiosis is a tick-borne disease caused by the intraerythrocytic apical complex parasite of the genus *Babesia* [1]. The majority of the pathogens are *Babesia microti, B. duncani*, and *Babesia divergens* [1]. Infection can result in fever, hemolytic anemia, and in severe cases, respiratory distress, pulmonary edema, and even death [2]. Patients who are immunocompromised or have undergone splenectomy are more susceptible to the disease and have a higher mortality rate [3], indicating that *Babesia* is a serious health risk for humans.

Understanding *Babesia* growth in the erythrocytes is critically important to mitigate babesiosis. The technique of cultivating *Babesia* is developed for animal species. Cattle parasite *Babesia bovis* was the first one to be established in continuous cultures using an agitated suspension culture technique and later on using a stationary layer of erythrocytes called the microaerophilous stationary phase (MASP) culture, where the parasites proliferate in a settled layer of blood cells [4]. Subsequently, more in vitro culture was established in *Babesia* species including *B. divergens* [5] and *Babesia bigemina* [6]. In 1982, *B. divergens* passaged several times in splenectomized calves and were cultured continuously for 85 days in a medium of 60:40 M199 and normal bovine serum at pH 7.2–7.4 [5]. In 1985, a *B. bigemina* strain was isolated from an infected calf and cultured continuously in vitro for more than 99 days in M199 (containing 20% to 50% fresh normal bovine serum) medium with 5% to 10% (*v*/*v*) erythrocyte suspension and 5% CO_2_, 2% O_2_, 93% N_2_ [6].

*B. duncani* (WA1) continuous in vitro culture was first reported in 1994, M199 medium supplemented with 40% FBS and blood from a low parasitemic hamster (3.5%) were used to establish the culture [7]. After cryopreservation and resuscitation culture, *B. duncani* showed that pathogenicity was maintained through hamster inoculation. Subsequently, the cultivation method of *B. duncani* has been continuously improved. HL-1 medium is widely used to culture *Babesia*. HL-1 is a serum-free, chemically defined medium for the in vitro culture of hybridomas and other lymphoid-derived cells [8]. HL-1 medium was initially used successfully in the in vitro culture of *Theileria equi* [8] and *Babesia caballi* [9]. *B. duncani* showed sustained proliferation in HL-1 medium supplemented with 20% FBS using hamster blood [10]. This combination of HL-1 medium supplemented with 20% FBS was also successfully used in the long-term in vitro culture of *B. duncani* with human red blood cells (hRBCs) [11]. Claycomb medium was well used to culture HL-1 cells from the AT-1 mouse atrial cardiomyocyte tumor lineage, allowing continuous passaging of HL-1 cells and maintaining the contracted, differentiated phenotype of the cells [12]. Subsequently, Claycomb medium was also used for the in vitro culture of *B. duncani* in hRBCs [11]. Recently, the replacement of HL-1 medium with DMEM/F12 medium also supported long-term in vitro culture of *B. duncani* in hRBCs, and proliferation was similar to the level in HL-1 medium [13]. However, a high concentration of serum still needs to be added to the medium to maintain the continuous growth of the parasites in vitro, and none of the media used so far for the in vitro culture of *B. duncani* can achieve serum-free culture.

The high concentration of serum in the culture medium hinders the purification of the antigens secreted by the parasite or the identification of the components required for the growth of the parasite in vitro, therefore it is particularly important to develop serum-free culture systems for *Babesia*. *B. divergens* was first achieved in serum-free in vitro culture [14,15]. The addition of a lipid carrier (AlbuMax^TM^ I or bovine serum albumin-Cohn fraction V) to the basal medium RPMI 1640 resulted in a PPE of 30% for *B. divergens*, close to that obtained in RPMI 1640 supplemented with 10 % human serum [15]. HL-1 medium supplemented with lipid-rich bovine serum albumin (LR-BSA) alone or different amounts of LR-BSA and chemically defined lipids (CDL) supported the in vitro serum-free proliferation of *B. caballi*, and the growth of *B. caballi* was maintained for more than 6 months [16]. A versatile medium, GIT medium was also used for the serum-free in vitro culture of *B. caballi* [17] and *B. bovis* [18]. Recently, a culture medium without animal components, VP-SFM, allowed the successful serum-free growth of *B. bigemina* [6] and *B. bovis* [19].

Fatty acids and lipids have been shown to provide energy, as well as cell membrane components involved in transport and signaling pathways, playing an important role in media without bovine serum supplementation [20]. Both AlbuMax^TM^ I and CD lipid mixture are lipid-rich supplements [6,15]. In the field of *Babesia*, AlbuMax^TM^ I was substituted for human serum to successfully achieved long-term serum-free in vitro culture of *B. divergens* in 1996 [15]. Recently, the CD lipid mixture added to VP-SFM was shown to maintain the long-term serum-free in vitro proliferation of *B. bigemina* [17] and *B. bovis* [18]. The presence of serum during in vitro culture has severely limited the studies of *B. duncani* in the fields of physiological function and vaccine development, and the expensive serum is a relatively large economic burden. Therefore, the development of in vitro serum-free cultures is essential for the study of *B. duncani*.

In this study, we investigated whether VP-SFM medium supplemented with AlbuMax^TM^ I or CD lipid mixture would support the in vitro serum-free proliferation of *B. duncani*. We determined a long-term in vitro serum-free culture method for the proliferation of *B. duncani*. This study will greatly improve our understanding of human babesiosis and provide an excellent resource for the design of vaccines and drugs for the prevention and treatment of human babesiosis, and provide an excellent foundation for a new era of advanced *Babesia* research.

## 2. Materials and Methods

### 2.1. In Vitro Culture of B. duncani in Different Growth Media

Hamster blood was used as donor RBC. Hamsters were anesthetized with isoflurane for retro-orbital venipuncture. The Blood was collected with EDTA K2 (solution/RBCs = 1:9; 10% EDTA-2K). The blood was centrifuged at 500× *g* for 10 min, followed by three washes of the cells by using 5 volumes of PSG, with careful removal of the supernatant and buffy coat at each wash. Next, the washed RBC with an equal volume of PSG plus extra glucose (20 g glucose/L) with a final concentration of 200 μg/mL streptomycin and 200 U/mL penicillin, was stored at 4 °C for a maximum of 2 weeks.

In vitro culture of *B. duncani* using HL-1 medium (LONZA, Shanghai, China) as previously described [21]. In vitro serum-free cultures of *B. duncani* were performed by using the animal product-free VP-SFM AGT^TM^ (Gibco Life Technologies, Shanghai, China) as the basal medium. The AlbuMax^TM^ I (Gibco Life Technologies, Shanghai, China) or CD lipid mixtures (Gibco Life Technologies, Shanghai, China) were added to the medium. 200 mM L-glutamine (Sigma, Shanghai, China), 2% antibiotic/antimycotic 100 × (Corning, Shanghai, China) was added as supplements. The parasite growth at 37 °C in a microaerophilous stationary phase ((5% CO_2_, 2% O_2_, 93% N_2_).

### 2.2. Effect of VP-SFM with Different Concentrations of AlbuMax^TM^ I on the Proliferation of B. duncani

Four concentrations of AlbuMax^TM^ I, 0.5 mg/mL, 1 mg/mL, 2 mg/mL, and 4 mg/mL, were added to VP-SFM to be evaluated for the in vitro proliferation of *B. duncani*. The initial PPE of the culture was 0.5% and the proliferation of *B. duncani* was observed for 3 days. Thin blood smears were made from each well, fixed in methanol, and subjected to Giemsa staining to monitor parasite proliferation. 3000 erythrocytes were counted per smear and PPE was determined. Three separate experiments were performed using three wells for each test condition.

### 2.3. Effect of VP-SFM with Different Concentrations of CD Lipid Mixture on the Proliferation of B. duncani

A chemically defined CD lipid mixture was diluted into four dilutions of 1:50, 1:100, 1:200, and 1:400 to evaluate the effect on *B. duncani* proliferation by adding each dilution to VP-SFM. VP-SFMA was used as a control. The initial PPE of the culture was 0.5%. Parasitemia was monitored every 24 h for 3 days by light microscopy examination of Giemsa-stained blood smears. 3000 erythrocytes were counted per smear. Three separate experiments were performed using three wells for each test condition.

### 2.4. B. duncani Long-Term Culture by Using VP-SFMA or 1:100 CD Lipid Mixture with VP-SFM

To test if the medium could support long-term cultivation, both mediums were tested by using an initial PPE of 0.5%. Parasites were splited every three days. Parasitemia was monitored every 24 h by light microscopy examination of Giemsa-stained blood smears. Parasite forms of *B. duncani* cultured with VP-SFMA and 1:100 CD Lipid mixture with VP-SFM for one generation were counted. 3000 erythrocytes were counted per smear. Three separate experiments were performed using three wells for each test condition.

### 2.5. Comparison of B. duncani Proliferation between VP-SFMA and HL-1 + 20% FBS Medium

HL-1 supplemented with 20% FBS has been used for continuous in vitro culture of *B. duncani* since 2018 [10,11]. To test the new formula, parasite proliferation cultured in VP-SFMA and HL-1 + 20% FBS medium was monitored daily by using an initial PPE of 0.5% for 3 days. Parasitemia was monitored every 24 h by light microscopy examination of Giemsa-stained blood smears. 3000 erythrocytes were counted per smear. Three separate experiments were performed using three wells for each test condition.

### 2.6. Virulence Assays in Syrian Golden Hamsters

To assess the virulence of *B. duncani* in serum-free culture, the parasites were propagated in vitro in hamsters’ red blood cells, and 10^7^ infected red blood cells were injected separately into female Syrian Golden hamsters (n = 3) by the intraperitoneal (IP) route. Blood smears were observed daily from day 2 after inoculation. Female Syrian Golden hamsters were briefly anesthetized with a respiratory anesthesia machine and blood was collected from the hind leg vein to prepare a thin blood smear. Parasitemia was monitored over time by light microscopic examination of Giemsa-stained blood smears. The PPE was determined by counting 3000 erythrocytes for each smear. Moribund Syrian golden hamsters were humanely euthanized.

### 2.7. Statistical Analysis

The data were analyzed using GraphPad Prism 7 (San Diego, CA, USA) by two-way analysis of variance (ANOVA), followed by Turkey’s multiple comparison test. Results are shown as mean + SD. NS, *p* > 0.05 not significant at 5%; *, *p* < 0.05 significant at 5%; **, *p* < 0.01 significant at 1%; ***, *p* < 0.001 significant at 0.1%. Error bars represent standard deviations.

## 3. Results

### 3.1. VP-SFM Medium with Different Concentrations of AlbuMax^TM^ I Affect the Proliferation of B. duncani In Vitro

VP-SFM with 2 mg/mL AlbuMax^TM^ I (VP-SFMA) was the most appropriate formula for in vitro growth of *B. duncani*, with PPE of 42 ± 2% at day 3 (Figure 1). While 4 mg/mL, 1 mg/mL, and 0.5 mg/mL AlbuMax^TM^ I had a maximum PPE of 34%, 25%, and 21% on day 3, respectively. Statistical analysis showed that the parasitemia of VP-SFMA is significantly higher than 4 mg/mL (*p* = 0.0048), 1 mg/mL (*p* = 0.0004), and 0.5 mg/mL (*p* = 0.0001), respectively.

### 3.2. Effect of VP-SFM with Different Concentrations of CD Lipid Mixture on the Proliferation of B. duncani

CD lipid mixture was suitable for serum-free culture as well in short term. The optimum dilution that maintained the in vitro growth of *B. duncani* was 1:100 with a maximum PPE of 36.5 ± 4.2% on day 3 (Figure 2). There is no significant difference in growth curves between the 1:100 CD lipid mixture compared to VP-SFMA (*p* = 0.5147). While the maximum PPE was 19.5% and 10.1% for the 1:200 and 1:400 CD lipid mixture, and only 0.3% for the 1:50 CD lipid mixture on day 3, respectively. Statistical analysis showed that the parasitemia of the 1:100 CD lipid mixture is significantly higher than 1:200 (*p* = 0.0040), 1:400 (*p* = 0.0008), and 1:50 (*p* = 0.0003), respectively.

### 3.3. B. duncani Long-Term Culture by Using VP-SFMA or CD Lipid Mixture

Both VP-SFMA and CD Lipid could support *B. duncani* growth, but the CD lipid mixture can only support 8 generations while VP-SFMA could support continuous long-term culture (Figure 3A). Parasites forms were also accounted for, the number of rings forms and tetrads forms increased significantly on day 3 compared to day 1 in both VP-SFMA (*p* = 0.0193, *p* = 0.0469), and 1:100 CD Lipid mixture (*p* = 0.0380, *p* = 0.0032) (Figure 3B). The filamentous forms were significantly lower on day 3 compared to day 1 in VP-SFMA (*p* = 0.0001) and 1:100 CD Lipid mixture (*p* = 0.0027) (Figure 3B). The proportion of infected red blood cells with rings, double rings, filamentous, and tetrads forms throughout the intraerythrocytic cycle was comparable in both media with no significant differences observed between the different developmental stages in the two media (Figure 3B).

### 3.4. Comparison of the Growth of B. duncani Culture by VP-SFMA and HL-1 Medium with 20% FBS

In comparison with HL-1 medium + 20% FBS, VP-SFMA had a better proliferation of *B. duncani*. VP-SFMA cultured *B. duncani* reached 41.5 ± 1% percentage of parasitized erythrocytes on day 3 (Figure 4). However, HL-1 medium with 20% FBS cultured *B. duncani* was only 22 ± 1% of the percentage of parasitized erythrocytes on day 3 (Figure 4). Based on parasitemia values, there was a significant difference between VP-SFMA and HL-1 medium with 20% FBS (*p* < 0.0001).

### 3.5. Serum-Free Culture of B. duncani Maintains Virulence against Hamsters

*B. duncani* of serum-free culture could infect Syrian Golden hamsters by intraperitoneal. *B. duncani* was observed in blood smears obtained from hamsters on day 2 post-inoculation, and PPE was over 30% on day 7 post-inoculation (Figure 5A). Morphological analysis of *B. duncani* in the erythrocytes of infected hamsters at a 10% PPE on day 5 of infection revealed that the infected cells were rings, double rings, filamentous and tetrads, with the major parasite morphology as rings (Figure 5B). All hamsters developed severe clinical signs, including dyspnea and chills. All experimental animals were euthanized on the 7 days.

## 4. Discussion

This is the first report of the serum-free in vitro proliferation of *B. duncani* with VP-SFM medium supplemented with AlbuMax^TM^ I or CD lipid mixture. The results showed that *B. duncani* achieved continuous in vitro proliferation in VP-SFMA (Figure 3A) and that *B. duncani* cultured with VP-SFMA maintained the pathogenicity of the parasite (Figure 5) and caused severe symptoms in Syrian golden hamsters. The parasite morphology (rings, double rings, filamentous forms, and tetrads) was also observed during our culture, suggesting that *B. duncani* maintained normal growth and development during serum-free culture [11,13]. The increase in *B. duncani* PPE was followed by an increase in the ring forms, but a decrease in the number of filamentous forms (Figure 3B), indicating that the increase in the number of parasites led to an increase in erythrocyte consumption, which possibly resulted in a decrease in the nutritional supply of the parasites, which mostly remained in the ring form.

The development of a continuous in vitro culture system for parasites is a challenge, but important for achieving genetic modification, drug screening, and functional studies. *Plasmodium falciparum* [22] and *Plasmodium knowlesi* [23] are the only two species that achieved continuous blood stage in vitro culture, and among human *Babesia* species, only *B. divergens* [15] and *B. duncani* [10,11] have achieved continuously in vitro culture.

In 1991–1993, the first clinical cases of human babesiosis caused by *B. duncani* were reported in Washington State and California respectively [24,25]. During the period 1994–2022, researchers have successively developed and improved in vitro culture systems for *B. duncani*, allowing for more intensive studies of *B. duncani* [7,10,11,13].VP-SFM is a basal medium intended for the growth of cell lines, the manufacturer declares that it does not contain proteins, peptides, or components of animal or human origin. This medium has been successfully used for the replication of various viruses such as rabies virus, Japanese encephalitis virus, yellow fever virus, and other viruses. There is study showed that Vero cell-cultured *Toxoplasma gondii* in a serum-free medium was able to maintain the same pathogenicity as test mice after multiple passages [26]. Recently, the VP-SFM medium was used in the in vitro culture of *B. bigemina* [6] and *B. bovis* [19]. We also tried to replace the HL-1 medium with VP-SFM for serum-free culture of *B. duncani* and found that *B. duncani* was normally able to proliferate, with PPE reaching a maximum of about 20% at day 3. The long-term serum-free in vitro culture of *B. bigemina* and *B. bovis* using VP-SFM medium was considered that putrescine in VP-SFM might play an important role [6,19]. Additional addition of putrescine to *B. bigemina* and *B. bovis* using A-DMEM/F12 medium in the in vitro culture further improved the PPE [27]. Putrescine is a polyamine that has been shown to play an important role in the proliferation process in *B. bigemina* [27], *B. bovis* [28], and other apical complex parasites such as *Toxoplasma* [6], *Leishmania* [6], and *P. falciparum* [29]. In another study, it was reported that *T. gondii*, *Plasmodium falciparum*, *Trypanosoma brucei*, *Leishmania donovani,* and *B. bovis* cannot synthesize polyamines from the head pathway and need to obtain them from the external environment [29,30,31,32,33].

Previous reports have shown that lipid metabolism plays an important role in the growth and development of protozoan parasites [34]. *P. falciparum* can synthesize only small amounts of fatty acids during intraerythrocytic proliferation and needs to utilize large amounts of fatty acids from the host serum [33,35]. *Toxoplasma gondii* can synthesize most lipids from cleared host cell precursors, achieving de novo parasitic acyl-lipid synthesis and recycling of host cell compounds coexist [36]. AlbuMax^TM^ I is composed mainly of serum albumin and lipids [15]. Extracting serum albumin and lipids from AlbuMax^TM^ I and supplementing them separately to RPMI medium did not improve the growth of *B. divergens*, the addition of both together significantly improved the proliferation of *B. divergens* [15]. In this study, we successfully achieved in vitro proliferation of *B. duncani* with the combination of VP-SFMA (Figure 1). Subsequently, we compared this optimal concentration combination of VP-SFMA with HL-1 with 20% FBS (Figure 4). Results showed that VP-SFMA resulted in better in vitro proliferation of *B. duncani* (*p* < 0.0001). This suggests that AlbuMax^TM^ I, as a serum substitute, compensates for the nutrient uptake of FBS by *B. duncani*, and together with the rich nutrients in VP-SFM, results in a better proliferation of *B. duncani* by VP-SFMA. The CD lipid mixture was likewise used as a lipid-rich supplement, and given that the combination of VP-SFM medium and CD lipid mixture was successfully used in *B. bigemina* [6] and *B. bovis* [19]. In the present study, *B. duncani* can also achieve similar proliferation in VP-SFM with a 1:100 CD lipid mixture as in VP-SFMA (Figure 2). Excessive concentrations of CD lipid mixture inhibited the growth of *B. duncani*. However, long-term culture cannot be achieved, the exact cause is not known and requires further study.

The host nutrition dependence of *B. duncani* is similar to that of other intraerythrocytic parasites such as *P. falciparum* [37,38]. *P. falciparum* survival in human erythrocytes is dependent on nutrients such as carbohydrates, lipids, and vitamins, and the proteins and enzymes used by the parasites to uptake and utilize these nutrients are considered to be promising targets for the development of new antimalarial drugs [37,39,40,41]. Future efforts are needed to identify the nutrients required for *B. duncani* survival in erythrocytes and to understand their uptake and utilization mechanisms.

## 5. Conclusions

The addition of serum to culture media plays an important role in maintaining the growth of *Babesia* in vitro. However, there are problems with the variability of serum batches, the possible presence of contaminants, and the high cost of good-quality serum. In this study, we successfully established in vitro serum-free long-term culture of *B. duncani* with VP-SFMA and demonstrated that it remained virulent after infecting hamsters. This provided an important laboratory resource for the study of the physiological mechanisms of *B. duncani*, the testing, and development of anti-*Babesia* drugs, diagnostic testing, and vaccine development.

## Figures and Tables

**Figure 1 cells-12-00482-f001:**
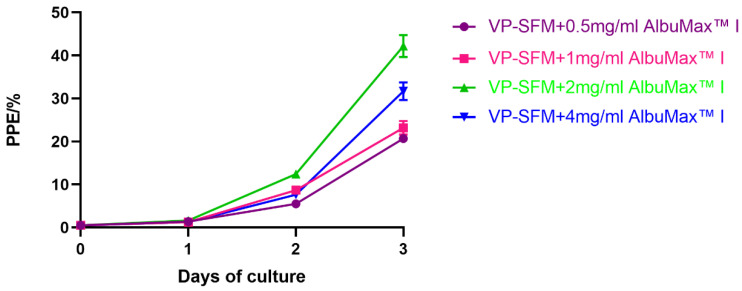
In vitro culture of *B. duncani* in VP-SFM supplemented with different concentrations of AlbuMax^TM^ I. Mean values from three separate experiments using triplicate wells for each test concentration were represented. PPE: percentage of parasitized erythrocytes.

**Figure 2 cells-12-00482-f002:**
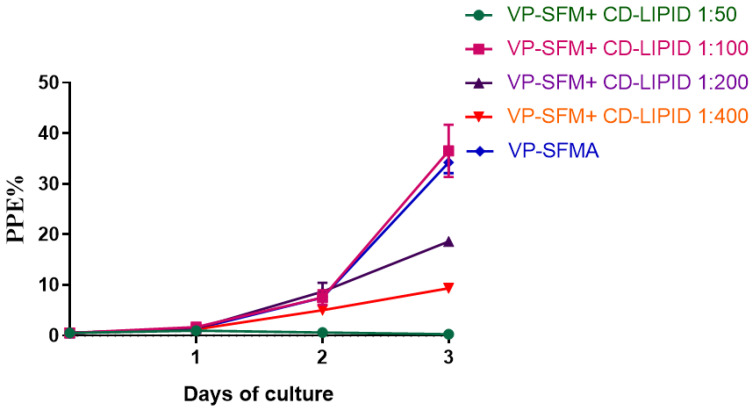
In vitro culture of *B. duncani* in VP-SFM supplemented with different concentrations of CD Lipid mixture. Mean values from three separate experiments using triplicate wells for each tested condition were represented.

**Figure 3 cells-12-00482-f003:**
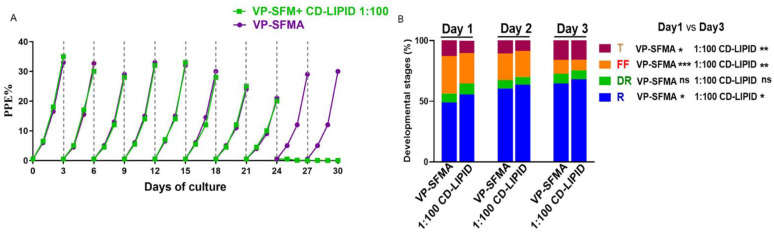
VP-SFMA supports long-term continuous in vitro culture of *B. duncani*. (**A**) Comparison of the growth curve of *B. duncani* proliferation using VP-SFMA and VP-SFM +CD LIPID (1:100) medium over a 30-day period with 9 splittings. Mean values from three separate experiments using triplicate wells for each tested condition were represented. PPE: percentage of parasitized erythrocytes. Dotted lines indicate subculturing performed every 3 days and adjusted back to 0.5%. (**B**) Percentages of different parasite development stages in VP-SFMA and 1:100 dilution of CD Lipid mixture. R, ring; DR, double rings; FF, filamentous, T, tetrads. ns: *p* > 0.05, *: *p* < 0.05, **: *p* < 0.01, ***: *p* < 0.001.

**Figure 4 cells-12-00482-f004:**
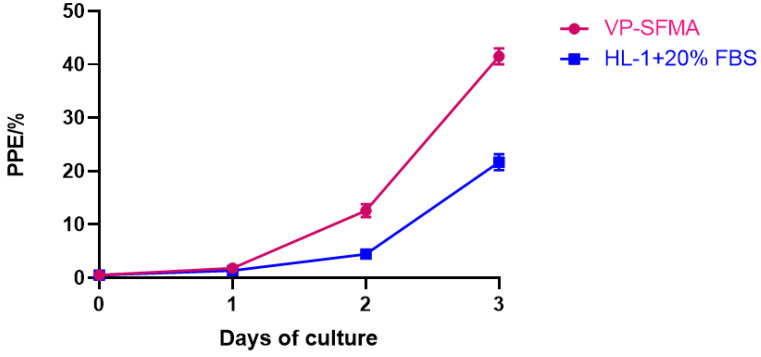
Comparison of growth curve of *B. duncani* proliferation using VP-SFMA with HL-1 + 20% FBS medium. Values represent the mean from three separate experiments using triplicate wells for each test condition. PPE: percentage of parasitized erythrocytes.

**Figure 5 cells-12-00482-f005:**
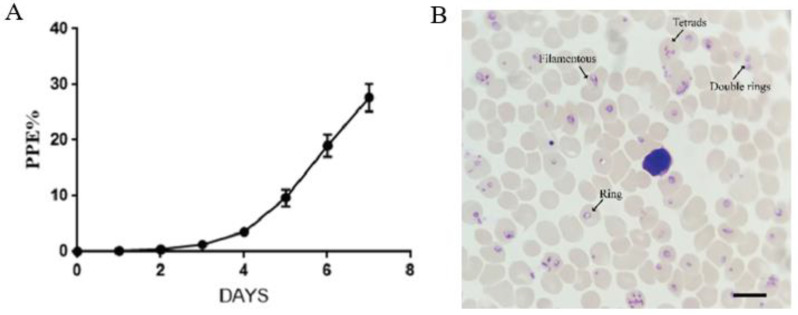
Serum-free culture of *B. duncani* infects Syrian Golden hamsters. (**A**) The PPE of *B. duncani* after intraperitoneal (n = 3). (**B**) Parasitic morphology of *B. duncani* in hamster erythrocytes in a Giemsa-stained smear prepared on day 5 after infection. The scale represents 10 μm.

## Data Availability

Not applicable.

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
