# Peer review of "Continuous In Vitro Culture of Babesia duncani in a Serum-Free Medium"

_cells, 2023, doi:10.3390/cells12030482_

Round 1

Reviewer 1 Report

Manuscript titled „Continuous in vitro culture of Babesia duncani in a serum-free medium” submitted by Weijun Jiang, Sen Wang, Dongfang Li, Yajun Zhang, Wanxin Luo, Junlong Zhao  and  Lan He presents interesting results on important tick-borne pathogen but unfortunately some issues are found.

Dear Authors, below is the list of flaws that should be revised in my opinion:

Abstract:

-       should be revised because of some minor linguistics issues and definitely abbreviations should be clearly explained to meet Journal requirements.

Introduction:

-  this section and the whole manuscript should be checked and all abbreviations should be explained (where they are used for the first time);

-  L40,41:  B. duncani, B. divergens should be used instead of full names; the whole manuscript should be traced to avoid this type of mistakes;

-     L50: mistake (empty place) in word microaerophilus;

-  L101-108:  paragraph should be revised - the aim of the study is mixed with results and conclusions;

Methods:

- The whole section should be revised and all missing details should be given;

-    L150 and others: “p” meaning significance level should be in Italics;

-   L129-131: give details about parasitemia monitoring; it is placed in Figure 2 description where it is not needed;

- L133-139: this paragraph should be revised. The Authors should give results of statistical tests where available and determine the level of statistical significance “p”, also should describe which data were normally distributed  and which not; and give the statistical software name etc.; Describe exactly which developmental stages would undergo analysis;

-     L173: unnecessary capital “P”;

Results:

-   whole section should be revised carefully to avoid describing methods or interpreting/explaining the results of present study in this section;

-    L143-145: this sentence belongs to Methods;

-    L153-154: these sentences belong to Methods;

-    L164-167: these sentences belong to Methods;

- L173-175: this sentence belongs to Discussion or should be revised if possible;

-    L181-183: This sentence belongs to Methods;

-    L191-193: this sentence belongs to Methods;

-   FIGURES: all figures should be placed in the nearest distance to their first citation;

-    L204-205: this sentence belongs to Methods;

-  Figure 3: The Authors should explain the levels of significance marked with asterisks in figure and in appropriate part of Methods;

Discussion:

-    L221: citation of  appropriate Figure/s in this sentence is needed; whole section should be checked and missing citations of Results (including Figures) should be improved;

-    L223: unnecessary space before word “rings”;

-    L234-236: is this sentence relevant in this paragraph?

-    L245-248: this sentence should be revised;

-     L263-268: this sentence should be revised;

-  L277-280: is this speculation or it is supported with the Authors own studies or previously published results? If it was confirmed then citation is required, if not then sentences should be revised;

-     L294: Babesia in Italics;

References: all Latin names should be in Italics;

I believe that after improvement of mentioned flaws this Manuscript should be considered for publication.

Reviewer 2 Report

The manuscript revealed that an in vitro serum-free long-term culture of B. duncani with VP-SFMA was successfully established and maintained virulence after infecting hamsters. Overall, in my opinion, it is an interesting manuscript However, there are some research gaps, which I have some points to mention below.

-Line 57: CO2, O2, N2, the number should be subscripted.

-For the results, Had the authors previously attempted to add the FBS in VP-SFM? I found it interesting that you compared HI1+20%FBS and VP-SFMA in Figure 4. Different concentrations of %FBS may stimulate or inhibit the growth of B. duncani. However, you can discuss this point.

-Line 220-223: This sentence needs to be rewritten because it was hard to understand.

-Check the word "in vitro" It italicized?

- The first word "PPE" should include the full name as shown in Figure 1.

- Add more method details to the section of the Virulence Assays in Mice.

- Do you have any pictures of the Serum-free culture of B. duncani maintaining virulence against hamsters? Please show and explain more detail in the results for easy-to-understand and valuable background information.

Round 2

Reviewer 2 Report

This manuscript can be published in this journal.
